# CRISPR/Cas9-Mediated Multi-Locus Promoter Engineering in *ery* Cluster to Improve Erythromycin Production in *Saccharopolyspora erythraea*

**DOI:** 10.3390/microorganisms11030623

**Published:** 2023-02-28

**Authors:** Xuemei Zhang, Yan Wang, Yue Zhang, Meng Wang

**Affiliations:** 1School of Life Sciences, Division of Life Sciences and Medicine, University of Science and Technology of China, Hefei 230026, China; 2Tianjin Institute of Industrial Biotechnology, Chinese Academy of Sciences, Tianjin 300308, China; 3Haihe Laboratory of Synthetic Biology, Tianjin 300308, China; 4Key Laboratory of Engineering Biology for Low-Carbon Manufacturing, Tianjin Institute of Industrial Biotechnology, Chinese Academy of Sciences, Tianjin 300308, China; 5Key Laboratory of Systems Microbial Biotechnology, Chinese Academy of Sciences, Tianjin 300308, China; 6College of Biotechnology, Tianjin University of Science and Technology, Tianjin 300457, China

**Keywords:** *Saccharopolyspora erythraea*, erythromycin, promoter engineering, CRISPR/Cas9, transcriptional level, expression fine-tuning, production

## Abstract

Erythromycins are a group of macrolide antibiotics produced by *Saccharopolyspora erythraea*. Erythromycin biosynthesis, which is a long pathway composed of a series of biochemical reactions, is precisely controlled by the type I polyketide synthases and accessary tailoring enzymes encoded by *ery* cluster. In the previous work, we have characterized that six genes representing extremely low transcription levels, *SACE_0716*-*SACE_0720* and *SACE_0731*, played important roles in limiting erythromycin biosynthesis in the wild-type strain *S. erythraea* NRRL 23338. In this study, to relieve the potential bottlenecks of erythromycin biosynthesis, we fine-tuned the expression of each key limiting *ery* gene by CRISPR/Cas9-mediated multi-locus promoter engineering. The native promoters were replaced with different heterologous ones of various strengths, generating ten engineered strains, whose erythromycin productions were 2.8- to 6.0-fold improved compared with that of the wild-type strain. Additionally, the optimal expression pattern of multiple rate-limiting genes and preferred engineering strategies of each locus for maximizing erythromycin yield were also summarized. Collectively, our work lays a foundation for the overall engineering of *ery* cluster to further improve erythromycin production. The experience of balancing multiple rate-limiting factors within a cluster is also promising to be applied in other actinomycetes to efficiently produce value-added natural products.

## 1. Introduction

Erythromycins, usually composed of erythromycin A, B, C, and D (ErA, ErB, ErC, ErD), is a group of clinically important macrolide antibiotics produced by *Saccharopolyspora erythraea*. Erythromycin biosynthesis is first catalyzed by the type I polyketide synthases (PKSs) encoded by the *ery* cluster and proceeds stepwise controlled by different enzymes: first, one molecule of propionyl-CoA and six molecules of methylmalonyl-CoA are assembled by EryA to form the 14-memberd macrolide 6-deoxyerythronolide B (dE-B); then two monosaccharides, *L*-mycarose and *D*-desosamine, are synthesized and added to the lactone ring by EryB and EryC (Appendix A), respectively, generating the intermediate ErD; at last, ErD undergoes post-modifications including hydroxylation (by EryK) and methylation (by EryG) to yield the final product ErA [1,2]. Like many other native antibiotic producers, the wild-type strain *S. erythraea* NRRL 23338 represents low erythromycin production. Secondary biosynthesis may be restricted in many aspects such as poor precursor supply, low expression of biosynthetic genes, and complex inherent regulation [3]. In the previous work, by the comparative transcriptional analysis of the high- and low-producing strains, we have demonstrated that the insufficient expression of *ery* genes is the main cause leading to the low production of the native producer and six genes representing extremely low transcription levels, *SACE_0716*, *SACE_0717*, *SACE_0718*, *SACE_0719*, *SACE_0720*, and *SACE_0731*, encoding EryCIV, EryBVI, EryCVI, EryBV, EryBIV, and EryBIII, respectively, were characterized to play important roles in limiting erythromycin biosynthesis in *S. erythraea* NRRL 23338 [4].

Eliminating the potential bottlenecks of a long pathway is the precondition for efficient secondary biosynthesis, where enhancing the expression of key limiting genes is the most direct and effective approach [3]. To this end, both gene overexpression and promoter engineering are commonly used strategies. However, introducing extra gene copies, either by genome integration or carried by the replicative plasmids, may induce metabolic burden to the host cells [5], and it is difficult to simultaneously overexpress multiple limiting genes due to the inaccessibility of enough selection markers and the high failure rate of large plasmid transformation. In addition, repetitive integration of the same gene may induce unnecessary genome instability. Promoter engineering is a practical way to fine-tune gene expression for metabolic engineering applications [6]. In contrast to traditional gene overexpression, in situ promoter engineering can enhance target gene expression by replacing the native promoter with a stronger one without introducing excess DNA sequences. Benefited from the fast-developing genetic toolkits, especially the successful applications of CRISPR techniques in actinomycetes [7,8,9,10,11], it becomes easier and more efficient to manipulate biosynthetic pathways in actinomycetes in a scarless manner, and multi-locus promoter substitution can be accomplished through CRISPR-mediated iterative genome editing and plasmid curing with the help of only one selection marker.

Although promoting gene expression is usually effective to enhance secondary biosynthesis, the expression of target genes needs to be adjusted to appropriate levels to not only supply sufficient biosynthetic enzymes but also avoid wasting intracellular energy and resources. Moreover, multiple rate-limiting factors may co-exist in a long pathway, so balancing the expression of several biosynthetic genes within a cluster is particularly crucial for promoting biosynthesis efficiency and maximizing yield of the target product [3]. Therefore, to achieve efficient secondary biosynthesis, it is meaningful to explore the optimal expression pattern of each target gene and provide a better approach for engineering multiple rate-limiting factors within a biosynthetic gene cluster.

In this study, to fine-tune the expression of six key limiting *ery* genes previously identified, we substituted their native promoters with three different heterologous ones of various strengths using the CRISPR/Cas9-mediated method and successfully improved erythromycin production of the engineered strains by 2.8- to 6.0-fold compared with the wild-type strain. Moreover, the optimal engineering strategies of each locus for maximizing erythromycin yield were also explored, revealing the coordination of synthases expression and product biosynthesis in different fermentation stages. Overall, our findings demonstrate the importance of balancing the expression of key limiting factors for efficient secondary biosynthesis, and they also provide valuable reference to in situ engineering secondary biosynthetic pathways for improved production of other important actinomycete-derived natural products.

## 2. Materials and Methods

### 2.1. Strains, Media and Cultivation Conditions

The strains used in this study are listed in Appendix A. *S. erythraea* NRRL 23338 was the parental strain for promoter engineering (accession number: NC_009142) [12]. *E. coli* DH5α was the recipient for plasmid construction. Plasmids were transferred into *S. erythraea* by conjugal transfer with the help of *E. coli* ET12567/pUZ8002 [13]. *E. coli* strains were cultivated in LB medium. R2YE liquid/agar medium and MS agar medium were used for *S. erythraea* strain cultivation and conjugal transfer, respectively. When necessary, antibiotics were supplemented as follows: 50 μg/mL of apramycin for the cultivation of both *S. erythraea* and *E. coli* strains, 80 and 200 μg/mL of hygromycin for the cultivation of *S. erythraea* and *E. coli* strains, respectively, 25 μg/mL of kanamycin and chloramphenicol for the cultivation of *E. coli* ET12567/pUZ8002, and 25 μg/mL of nalidixic acid for killing *E. coli* after conjugational transfer.

### 2.2. Fluorescence Detection and Microscopy

For fluorescence detection, *S. erythraea* eGFP expression strains were first cultivated in 2 mL R2YE liquid medium in a 24-well plate at 32 °C and 250 rpm for 24 h and then 200 μL seed broths were transferred into 2 mL R2YE liquid medium for another 18 h of cultivation at 32 °C and 250 rpm. Next, 100 μL broths were added into the 96-well ELISA plate (Corning Incorporated, Corning, NY, USA) for the detection of cell density and fluorescence (Ex 488 nm/Em 520 nm) using the microplate reader (BioTek Neo2, VT, USA). The relative fluorescence of each strain was normalized by the corresponding cell density value, and three biological replicates were analyzed to be accurate. Meanwhile, 20 μL mycelium-containing broth was used for observation using the fluorescence microscope under a 40× objective lens (Leica DM5000B, Wetzlar, Germany). The fluorescent photos were taken at an exposure time of 200 ms.

### 2.3. Construction of eGFP Expressing Strains

The plasmids constructed in this study are listed in Appendix A. Promoter sequences are listed in Appendix A. Four native promoters were amplified from the genomic DNA of *S. erythraea* NRRL 23338 genome. Three heterologous promoters and *egfp* gene were amplified from laboratory-stocked plasmids. To construct the eGFP expressing plasmids, promoter and *egfp* fragment were assembled with the integrative pSET152 backbone via homologous recombination. After PCR and Sanger sequencing verification, the successfully constructed plasmids were transferred into *S. erythraea* by conjugal transfer. Primers used for plasmid construction are listed in Appendix A.

### 2.4. CRISPR-Mediated Genome Editing

A 20 bp guide RNA (gRNA) for each target gene was calculated by the sgRNAcas9 tool [14], and the gRNA sequences are listed in Appendix A. To construct the CRISPR editing plasmids, the target heterologous promoter was fused with two 1 kb homologous arms (HA) in the flanks amplified from *S. erythraea* NRRL 23338 genome, generating the upHA-promoter-downHA fragment. Then the fragment was ligated to the replicative plasmid carrying *p_rpsL(XC)_*-Cas9-*t0* and *p_gapdh(EL)_*-sgRNA-*fd* cassettes and *pSG5* replicon, generating a series of CRISPR editing plasmids (Appendix A). CRISPR editing plasmids were transferred into *S. erythraea* NRRL 23338 by conjugal transfer. After successful genome editing, the temperature-sensitive CRISPR plasmids were removed by continuous cultivation for two generations at 39 °C. The strains, after successful plasmid curing, were verified by testing apramycin sensitivity on R2YE agar plates.

### 2.5. Evaluation of CRISPR Editing Efficiency

The emerging transformants after conjugal transfer were picked from MS agar plates and cultivated in R2YE liquid medium containing apramycin and nalidixic acid under 32 °C and 250 rpm. Then genomic DNA was extracted from three-day-old mycelium according to the manufacturer’s protocols (BIOMIGA bacterial gDNA isolation kit, Shanghai, China) and used as the template for PCR verification. The specific primer pair targeting heterologous promoter and the region outside the homologous arm was designed to recognize the positive colonies, and the PCR products were subjected to Sanger sequencing for further verification. CRISPR editing efficiency was calculated as follows: CRISPR editing efficiency = (Number of the positive colonies verified by PCR amplification and Sanger sequencing)/Number of the colonies survived in the R2YE liquid medium with antibiotics. Primers used for PCR verification are listed in Appendix A.

### 2.6. Real-Time Quantitative PCR Analysis

For RNA isolation, 1 mL fermentation broth after 3 and 7 days of cultivation, respectively, was used as the material for RNA isolation according to the manufacturer’s protocols (RNAprep pure cell/bacteria kit, TIANGEN, Beijing, China). Starting from the RNA template, cDNA was synthesized using the reverse transcription kit (ReverTra Ace qPCR RT Master Mix with gDNA Remover, TOYOBO, Osaka, Japan). Real-time quantitative PCR was performed in the thermal cycler (LightCycler 480II, Roche, Basel, Switzerland) using cDNA as the template according to the manufacturer’s protocols (SYBRGreen real time PCR Mix, TaKaRa, Osaka, Japan), where the housekeeping gene *sigA* (*SACE_1801*) was used as the internal reference. The relative expression level of each gene was calculated based on the comparative cycle threshold method [15]. The primers used for real-time quantitative PCR analysis are listed in Appendix A.

### 2.7. Fermentation and Determination of Erythromycin Production

Seven-day-old spores collected from the R2YE agar plate were inoculated into a 24-well plate containing 3 mL seed medium for 3 d, cultivated under 32 °C and 250 rpm. Then 300 μL seed broth was transferred into 3 mL fermentation medium for another 7 d fermentation under 32 °C and 250 rpm. The seed medium was prepared as follows: 10 g/L of glucose, 4 g/L of tryptone, 4 g/L of yeast extract, 0.5 g/L of MgSO_4_, 2 g/L of KH_2_PO_4_, and 4 g/L of K_2_HPO_4_. The fermentation medium was prepared as follows: 20 g/L of starch, 20 g/L of dextrin, 15 g/L of soybean powder, 4 g/L of (NH_4_)_2_SO_4_, 6 g/L of CaCO_3_, and 5 mL/L of soybean oil. At the end of fermentation, the remaining volume of the fermentation broth was first measured, and then the broth was mixed with an equal volume of ethyl acetate for extraction twice. The extract was concentrated in vacuo until the organic solvent was completely evaporated. The residue was dissolved in 300 μL acetonitrile and filtered by 0.22 μm membrane, generating the sample for further analysis. Erythromycin production was determined using the high performance liquid chromatography (HPLC) instrument (Agilent 1260 Infinity II, Santa Clara, CA, USA) and the C_18_ column (4.6 mm × 150 mm, 4 μm) (InfinityLab Poroshell 120 EC-C18, Agilent, Santa Clara, CA, USA). The column temperature was set at 35 °C. The analysis for each sample lasted 30 min where the mobile phases were 55% K_2_HPO_4_ solution (phase A) and 45% acetonitrile (phase B) at a flow rate of 0.8 mL/min. The UV signals were detected at 215 nm. The HPLC standard curves of ErA and ErB are provided in Appendix A.

### 2.8. Liquid Chromatography Mass Spectrometry (LC-MS) Analysis

LC-MS analysis was performed using Bruker Metabolic Profiler (HPLC (Agilent 1200, Santa Clara, CA, USA)-MS (Bruker micrOTOF-Q II, Karlsruhe, Germany)) equipped with the C_18_ column (4.6 mm × 150 mm, 4 μm) (InfinityLab Poroshell 120 EC-C18, Agilent, Santa Clara, CA, USA). The samples were prepared as previously described and were eluted by the mixture of 0.1% formic acid solution (phase A) and acetonitrile (phase B) at a flow rate of 0.8 mL/min with the linear gradient as follows: 90–5% of A for 30 min, 5% of A for 9 min, 5–90% of A for 1 min, and 90% of A for 10 min. Erythromycin and intermediates were detected by MS in the positive mode.

## 3. Results

### 3.1. Characterization of Native ery Promoters of Key Limiting Genes

In our previous study, six *ery* genes, *SACE_0716*, *SACE_0717*, *SACE_0718*, *SACE_0719*, *SACE_0720*, and *SACE_0731*, were demonstrated to play important roles in erythromycin biosynthesis, and overexpression of any one of the six genes led to improved erythromycin production in various degrees. In the *ery* gene cluster, these six genes were distributed in four transcriptional units, where *SACE_0716*-*SACE_0717* and *SACE_0719*-*SACE_0720* were co-transcribed by sharing 4 bp overlaps between the adjacent genes (Figure 1A). Four potential native promoters were located in the intergenic regions before *SACE_0717*, *SACE_0718*, *SACE_0720*, and *SACE_0731*, whose intact lengths were 84, 52, 224, 189 bp, respectively (Appendix A). All these regions exhibited low promoter activity in the calculation of the PromoterHunter tool [16]. To characterize the strength of the four native promoters, *p_ery18-17_*, *p_ery19-18_*, *p_ery21-20_*, and *p_ery32-31_*, the complete intergenic sequences were regarded as the potential promoters and amplified to drive the expression of the eGFP reporter. The eGFP expressing cassette was integrated to *S. erythraea* NRRL 23338 genome, and the fluorescence normalized by the biomass of each recombinant strain was used to evaluate the relative strength of the corresponding promoter. The native-promoter-harboring strains were observed to exhibit no significant difference in fluorescence compared with the wild-type strain, suggesting that all of the four native promoters represented extremely low activity in *S. erythraea* NRRL 23338, which would probably cause insufficient transcription of the potential key limiting genes in erythromycin biosynthesis (Figure 1B,C).

To enhance transcription of these six genes, three heterologous promoters of different strengths were selected as candidates for further in situ promoter substitution in four loci, among which *p_kasO_* is a well-known strong promoter used in many *Streptomyces* strains [17,18,19], and the other two promoters, *p_ermE*_s23_* and *p_2101_s32_*, derived from *p_ermE*_* and *p_SACE_2101_*, respectively, were synthetic promoters characterized in our previous study [4] (Appendix A). The strengths of these three heterologous promoters were also evaluated using the eGFP reporter system, and around 1.3-, 4.3-, and 13.3-fold higher GFP fluorescence was observed from the *p_2101_s32_*-, *p_ermE*_s23_*-, and *p_kasO_*-harboring strains, respectively, than the average level of the strains carrying native ones (Figure 1B), demonstrating the varied and stronger activities of the heterologous promoters. The microscopic observation result of eGFP expressing strains also demonstrated the significantly stronger activities of the heterologous promoters compared to the native ones, especially *p_kasO_* (Figure 1C). The selected heterologous promoters exhibited distinctive gradient of activities from each other, which will facilitate in situ expression fine-tuning of multiple key limiting genes in the following experiments.

### 3.2. CRISPR/Cas9-Mediated Promoter Engineering in ery Cluster

Gene overexpression by φ31 integration or the multi-copy replicative plasmid may cause excess metabolic burden to the chassis cell, which was particularly difficult in the manipulation of multiple target genes. To enhance gene expression in a scarless manner, CRISPR/Cas9 genome editing method was used to facilitate promoter substitution or knock-in at each locus. The CRISPR plasmid carried *p_rpsL(XC)_*-Cas9 and *p_gapdh(EL)_*-sgRNA expression cassettes, and the heterologous promoter was flanked by around 1 kb homologous arms (Figure 2A). Since *SACE_0720* and *SACE_0731* are transcribed divergently from their adjacent genes, the intergenic sequences between *SACE_0720*-*SACE_0721* and *SACE_0731*-*SACE_0732* probably process promoter activity in both directions. In order to maintain the transcription of the adjacent genes, heterologous promoters were knocked-in before *SACE_0720* and *SACE_0731*, while, in the other two loci, the native promoters were intactly replaced by the heterologous ones. After transformation, genome editing was successfully observed in all tested loci in the corresponding recombinant strains (Appendix A), and the editing efficiencies were determined to be 23.5%, 42.3%, 66.7%, and 50.0% at loci L1, L2, L3, and L4, respectively (Figure 2B).

To determine the optimal expression pattern of six key limiting genes for the most efficient erythromycin biosynthesis, three heterologous promoters were alternatively assembled in four loci using the CRISPR/Cas9 method, generating ten engineered strains, except that *p_kasO_* and *p_2101_s32_* failed to be knocked-in in the *SACE_0718* (L2) and *SACE_0720* (L3) loci, respectively (Figure 2C).

### 3.3. Improving Erythromycin Production by Promoter Engineering of Multiple ery Genes

To evaluate the performance of the engineered strains harboring different heterologous promoters, ten engineered strains and the wild-type strain *S. erythraea* NRRL 23338 were fermented in the 24-well plates, and the total production of ErA and ErB, which were the two dominant components in the fermentation extract, were determined by HPLC. Eight engineered strains, except SE/0720(*p_kasO_*) and SE/0731(*p_ermE*_s23_*), represented 2.8- to 6.0-fold improved erythromycin production compared with the wild-type strain, demonstrating the key limiting roles of overexpressed *ery* genes in erythromycin biosynthesis. Strain SE/0720(*p_ermE*_s23_*) produced 298.8 mg/L of total erythromycin, which was the highest production in all tested strains (Figure 3A). For different promoter engineering loci, the strains with higher productions were SE/0717(*p_2101_s32_*), SE/0718(*p_ermE*_s23_*), SE/0720(*p_ermE*_s23_*), and SE/0731(*p_kasO_*) for L1, L2, L3, and L4, respectively, albeit similar productions were also observed in strains SE/0717(*p_kasO_*) (L1), SE/0718(*p_2101_s32_*) (L2), and SE/0731(*p_2101_s32_*) (L4). Interestingly, strains SE/0720(*p_ermE*_s23_*) and SE/0720(*p_kasO_*), integrated with promoters *p_ermE*_s23_* and *p_kasO_* before *SACE_0720*, respectively, exhibited dramatically different productions, and the weaker promoter *p_ermE*_s23_* contributed to the higher production of the corresponding engineered strain (Figure 3A). The fermentation results of different promoter-engineered strains indicated that the strong promoter was not always optimal in gene expression for secondary biosynthesis. For example, for locus L2, promoters *p_2101_s32_* and *p_ermE*_s23_* with different strengths did not induce significant variation in biosynthesis efficiency. While, for locus L3, the weak promoter *p_ermE*_s23_* led to much higher production than the strong promoter *p_kasO_*. These results demonstrated the importance of expression fine-tuning of target genes, which would be beneficial to maximize production of target products.

In erythromycin biosynthesis, EryB enzymes were responsible for catalyzing erythronolide B (EB) to yield 3-α-mycarosylerythronolide B (MEB), which was consequently converted to erythromycin D (ErD) by EryC enzymes (Figure 3B). Altered expression levels of EryB and EryC enzymes might lead to re-distribution of the three related intermediates EB, MEB, and ErD and, consequently, influence the overall erythromycin biosynthesis and final production. To examine the effect of EryB and EryC overexpression in the engineered strains, the contents of EB, MEB, and ErD in the fermentation extract of *S. erythraea* NRRL 23338 and engineered strains were evaluated by mass spectroscopy. The results showed that EB and MEB were present in relatively low concentrations in almost all tested strains compared with ErD (Figure 3C). Significantly, in EryCIV and EryCVI overexpressing strains SE/0717(*p_2101_s32_*), SE/0717(*p_ermE*_s23_*), SE/0717(*p_kasO_*), SE/0718(*p_2101_s32_*), and SE/0718(*p_ermE*_s23_*), ErD accumulation was enhanced compared to that in the wild-type strain *S. erythraea* NRRL 23338 and in the other engineered strains, which was probably due to the improved expression of EryCIV and EryCVI (Figure 3C). Overall, the redistribution of intermediates in the engineered strains indicated the effective expression tuning of target key enzymes, which would probably alleviate the biosynthetic bottleneck and consequently improve erythromycin production.

### 3.4. Transcriptional Analysis of Key ery Genes in the Engineered Strains

Since engineering with varied promoters in the same locus might result in significantly different productions of the engineered strains, we were curious about the correlation between *ery* gene expression and erythromycin production. To evaluate the performance of three introduced promoters, we selected two series of engineered strains successfully integrated with three different heterologous promoters, *p_2101_s32_*, *p_ermE*_s23_*, and *p_kasO_*, to conduct the real-time quantitative PCR experiments, where the housekeeping gene *sigA* (*SACE_1801*) encoding the RNA polymerase sigma factor was used as the internal reference. The relative expression levels of the related target genes, *SACE_0716* and *SACE_0717* in strains SE/0717(*p_2101_s32_*), SE/0717(*p_ermE*_s23_*), and SE/0717(*p_kasO_*), and *SACE_0731* in strains SE/0731(*p_2101_s32_*), SE/0731(*p_ermE*_s23_*), and SE/0731(*p_kasO_*), were determined compared to the wild-type strain *S. erythraea* NRRL 23338. For the three *SACE_0716* and *SACE_0717* overexpressing strains, since *SACE_0716* and *SACE_0717* were co-transcribed, their expression levels changed synchronously in the whole fermentation process (Figure 4A). On day 3, gene expression was significantly improved by promoter substitution, and transcription attenuation was observed in all engineered strains as the fermentation time was prolonged to 7 days (Figure 4A). Interestingly, *p_2101_s32_*, which was mutated from the promoter of the housekeeping gene *SACE_2101*, exhibited stronger activity in the early fermentation stage (day 3) than the other two synthetic promoters, and weakened sharply in the late fermentation stage (day 7) (Figure 4A). Erythromycin productions of strains SE/0717(*p_2101_s32_*), SE/0717(*p_ermE*_s23_*), and SE/0717(*p_kasO_*) were well-correlated with the ranking of expression levels of *SACE_0716* and *SACE_0717* on day 3 (Figure 3A and Figure 4A), suggesting that sufficient expression of key limiting genes at the early stage might assert more influence to efficient secondary biosynthesis than that in the late stage.

We also noticed that, for some unknown reason, *SACE_0731* expression level did not change after promoter substitution in strain SE/0731(*p_ermE*_s23_*) on day 3 (Figure 4B). In fact, strain SE/0731(*p_ermE*_s23_*) produced 57 mg/L of erythromycin, which was similar to the production of the wild-type strain (Figure 3A), and the unknown mutations on the genome might be the reason for the failure of production improvement in this strain. This result again demonstrated that erythromycin production was strongly correlated with the actual expression levels of key limiting genes.

## 4. Discussion

Generally, secondary biosynthesis is a long biochemical reaction chain where a series of enzymes with orderly function cooperate precisely to synthesize the final product. In this process, one or multiple rate-limiting factors may exist and reduce the overall biosynthetic efficiency, which is often due to the insufficient expression of specific genes. It is not surprising that high producers of secondary metabolites compared to the wild-type strains demonstrated higher transcription levels of genes in biosynthetic cluster and those of pathway-related functions such as precursor supply and product efflux [20,21,22]. In strain engineering, the expression level of the target gene can be enhanced by introducing extra gene expression cassettes or using a stronger promoter to drive gene transcription to overcome the biosynthetic bottleneck and improve product yield [23,24,25]. Nevertheless, it is also worth noting that excess gene expression may lead to accumulation of toxic intermediates, which is a waste of intracellular resources and causes metabolic burdens that are harmful or even lethal to the host strain [5,26]. Thus, fine-tuning the expression of key biosynthetic genes, especially coordinating the expression of multiple rate-limiting factors within a pathway, is crucial to promoting biosynthetic efficiency and maximizing product yield. In this study, we explored the optimal expression pattern of multiple limiting genes in erythromycin biosynthesis by replacing their native promoters with heterologous ones of different strengths, and we constructed a series of engineered strains with improved erythromycin production via promoter engineering at four different loci. Of note, the required expression levels of individual limiting genes may differ significantly for the overall consideration of efficient secondary biosynthesis, and thus the best option for promoter engineering may strikingly vary with different engineering targets. In the engineering of the *SACE_0720* target, the integration of the weaker promoter *p_ermE*_s23_* resulted in 6.0-fold improved erythromycin, verifying the rate-limiting role of *SACE_0720* in erythromycin biosynthesis, while the substitution of the stronger promoter *p_kasO_* had no effect on production improvement, which we speculated was due to the expression imbalance of *ery* biosynthetic genes and the metabolic burden caused by the strong promoter.

In our previous work, we have identified that six *eryB* and *eryC* genes were low-expressing in the wild-type strain compared to the high producer [4], and their key limiting roles were verified by the results of gene overexpression and promoter engineering. It is reported that, in erythromycin biosynthesis, the formation rates of the intermediates MEB and ErD are lower than that of EB, which suggests that the whole pathway is limited by the synthesis and addition of two monosaccharides catalyzed by EryB and EryC enzymes [1]. However, we did not observe high concentrations of EB and MEB in either the wild-type strain or the engineered strains, but both significantly enhanced accumulation of ErD, and improved yields of the final product were detected in the *eryC*-overexpressing strains, which probably suggests better-matched formation rates of the macrolide and sugars. Tuning the expression of these limiting genes contributed to significant improvement of erythromycin production, indicating an alleviated bottleneck of sugar biosynthesis and more efficient secondary biosynthesis. With the help of CRISPR technology, we successfully performed promoter engineering at four individual loci in *S. erythraea* NRRL 23338 and determined the optimal promoter engineering strategy of each target. The CRISPR/Cas9 plasmid harboring *pSG5* replicon can be easily cured through high temperature cultivation and applied to the genome editing at another locus in the next round. Thus, these promoter engineering strategies can be rationally combined to achieve multi-locus promoter substitution through CRISPR-mediated iterative genome editing to simultaneously coordinate the expression levels of multiple factors, and the resultant engineered strain will likely perform better in the production of erythromycin.

In addition to the engineering efforts targeting the biosynthetic genes, there are also many successful examples of manipulating genes outside the *ery* cluster to improve erythromycin biosynthesis in *S. erythraea* strains. For rational strain engineering, enhancing precursor supply/energy availability [27,28,29], manipulating regulators/two-component systems [30,31,32], and overexpressing heterologous non-PKS genes such as the hemoglobin gene *vhb* [33], are proved to be effective strategies to enhance erythromycin production in *S. erythraea* to various degrees. The effectiveness of these modifications collectively suggests that erythromycin biosynthesis is probably restricted by a series of complex factors, which are not only located in the *ery* cluster but also distributed among the bacterial genome. The comparative genome analysis also revealed the wide distribution of gene mutations of the erythromycin hyper producers compared with the wild-type *S. erythraea* strain [21,22]. Therefore, to further improve erythromycin production, it is promising to combine the potential beneficial targets in and outside the gene cluster together to engineer strain, or to express the genetically modified biosynthetic gene cluster in an industrially optimized strain for an alternative [34].

## 5. Conclusions

To improve erythromycin production, rational strain engineering focusing on multiple biosynthetic genes is poorly documented compared with the larger number of studies in other aspects. However, synthases encoded by the biosynthetic genes are directly responsible for product formation, which will play dominant roles in the efficient proceedings of secondary biosynthesis. In this study, we fine-tuned the expression levels of six key limiting *ery* genes by promoter engineering at four different loci and determined the optimal engineering strategy of each locus using a series of heterologous promoters with a gradient of activities. Ten engineered *S. erythraea* strains were constructed for fermentation and transcriptional analysis, and their erythromycin productions were successfully improved by 2.8- to 6.0-fold compared to the wild-type strain. We also demonstrated that the appropriate expression of rate-limiting genes, which may significantly vary among targets, is crucial for maximizing erythromycin production. To the best of our knowledge, this work is the first to focus on the expression fine-tuning of four different loci in the *ery* biosynthetic cluster where six key *ery* genes are involved. This work is strongly complementary to previous engineering efforts in other aspects and provides a basis for the overall strain engineering through multiple approaches to further improve erythromycin production. In addition, the attempts to enhance secondary biosynthesis by clarifying the optimal expression pattern of multiple rate-limiting factors with catalytic functions are also applicable to not only in situ engineering but also *de novo* refactoring of other secondary biosynthetic pathways in actinomycetes, especially for those lacking pathway-specific regulators such as the *ery* cluster. Therefore, this work is of great significance for the efficient biosynthesis of other actinomycete-derived value-added natural products.

## Figures and Tables

**Figure 1 microorganisms-11-00623-f001:**
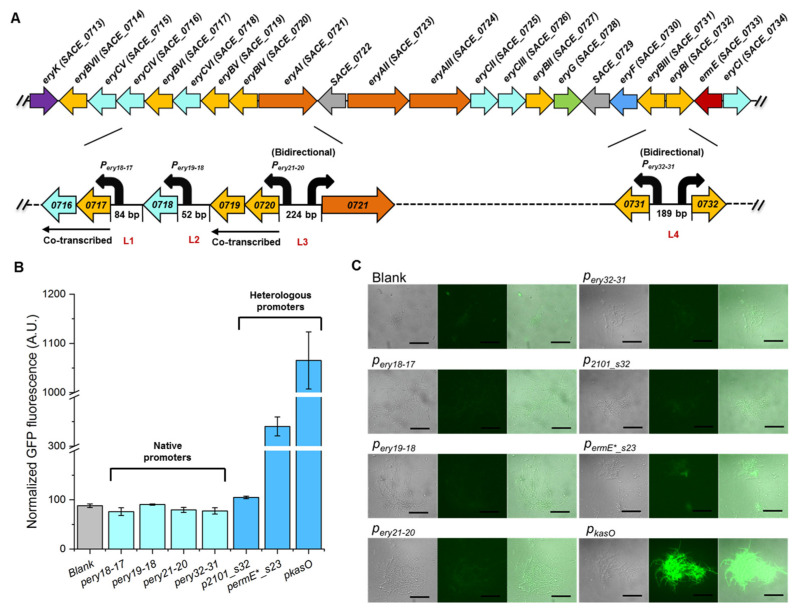
Activity determination of different promoters. (**A**) *Ery* biosynthetic gene cluster. Four native promoters are indicated by black arrows. Four loci for promoter engineering are indicated in red characters. (**B**) Relative strengths of native and heterologous promoters. The normalized fluorescence of the wild-type *S. erythraea* NRRL23338 was used as the blank control. Error bars represent standard deviations calculated from three independent biological replicates. (**C**) Microscopy of eGFP expressing strains harboring different promoters. Scale bar: 50 μM.

**Figure 2 microorganisms-11-00623-f002:**
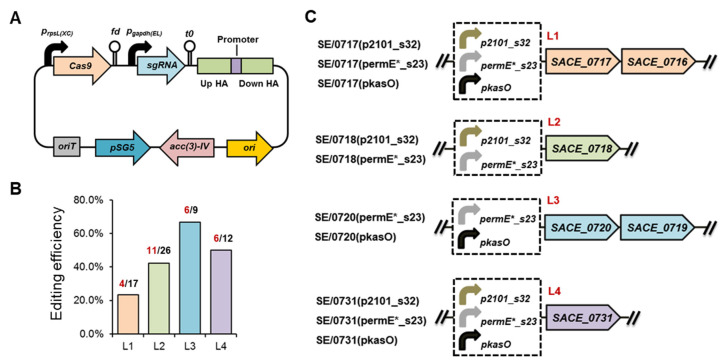
CRISPR/Cas9-mediated multi-locus promoter engineering in *S. erythraea* NRRL 23338. (**A**) Illustration of the CRISPR/Cas9 plasmid for genome editing. (**B**) CRISPR/Cas9 editing efficiencies at different loci. Numbers of correct colonies and verified colonies of each locus are shown in red and black, respectively. (**C**) Promoter engineering strategies at loci *SACE_0717* (L1), *SACE_0718* (L2), *SACE_0720* (L3), and *SACE_0731* (L4) and the names of the corresponding engineered strains.

**Figure 3 microorganisms-11-00623-f003:**
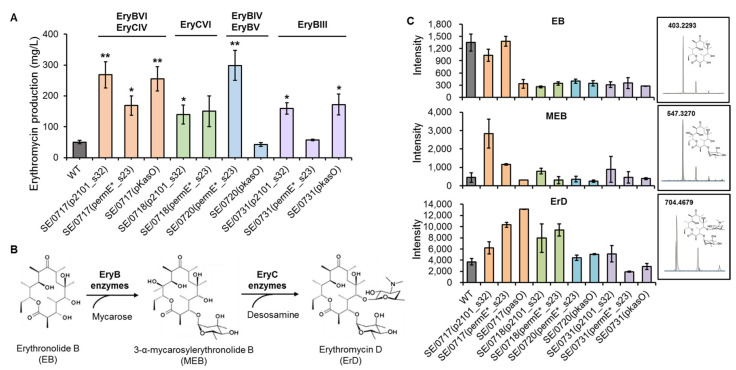
Erythromycin production analysis of engineered *S. erythraea* strains. (**A**) Total erythromycin production (ErA + ErB) of the wild-type strain *S. erythraea* NRRL 23338 (WT) and different engineered strains. The target enzymes of promoter engineering in each strain are indicated above the corresponding column. Error bars represent the standard deviation of three biological samples. * *p* ≤ 0.05 and ** *p* ≤ 0.01 (Student’ s two-tailed *t*-test). (**B**) Biosynthesis of the compounds related to EryB and EryC enzymes including erythronolide B (EB), 3-α-mycarosylerythronolide B (MEB) and erythromycin D (ErD). (**C**) Peak intensities of the intermediates EB, MEB, and ErD detected by MS in the fermentation extract of *S. erythraea* NRRL 23338 (WT) and engineered strains. The primary *m/z* value and MS spectrum of each compound are indicated in the box. Error bars indicate the standard deviations of three biological replicates.

**Figure 4 microorganisms-11-00623-f004:**
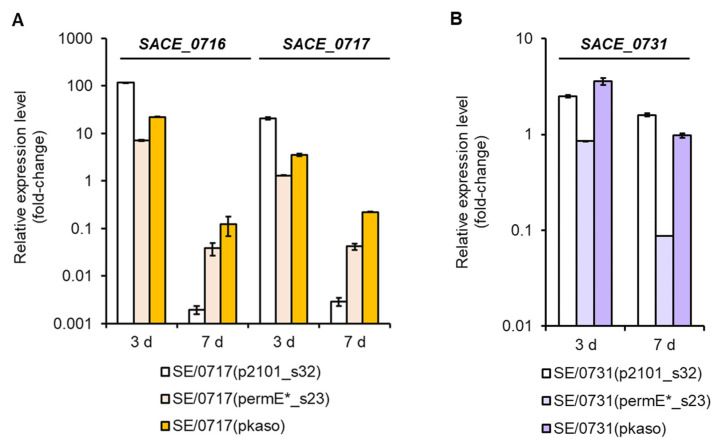
Relative expression levels of the target genes *SACE_0716*, *SACE_0717* (**A**), and *SACE_0731* (**B**) on day 3 and day 7 in the engineered strains harboring three heterologous promoters *p_2101_s32_*, *p_ermE*_s23_*, and *p_kasO_* compared to the wild-type strain *S. erythraea* NRRL 23338. The vertical *axis* was the logarithmic *axis*. Error bars represented the standard errors of the means (*n* = 3).

## Data Availability

The data presented in this study are available in this article and Appendix A.

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
