# Peer review of "CRISPR/Cas9-Mediated Multi-Locus Promoter Engineering in ery Cluster to Improve Erythromycin Production in Saccharopolyspora erythraea"

_microorganisms, 2023, doi:10.3390/microorganisms11030623_

Round 1

Reviewer 1 Report

Xuemei Zhang and co-authors present a quality and well-written experimental manuscript describing CRISPR/Cas9-mediated multi-loci promoter engineering in ery cluster to improve erythromycin production in Saccharopolyspora erythraea.

Authors fine-tuned the expression of each key limiting ery gene by CRISPR/Cas9-mediated multi-loci promoter engineering. The native promoters were replaced with different heterologous ones of various strengths, generating ten engineered strains, whose erythromycin productions were 2.8- to 6.0-fold improved compared to that of the wild-type strain. Additionally, the optimal expression pattern of multiple rate-limiting genes and preferred engineering strategies of each locus for maximizing erythromycin yield were also summarized. Their work lays a foundation for the overall engineering of ery cluster to further improve erythromycin production. The experience of balancing multiple rate-limiting factors within a cluster is also promising to be applied in other actinomycetes to efficiently produce value-added natural products.

Authors optimized the expression of previously identified six key limiting ery genes, they substituted their native promoters with three different heterologous ones of various strengths using the CRISPR/Cas9-mediated method, and successfully improved erythromycin production of the engineered strains. Moreover, the optimal engineering strategies of each locus for maximizing erythromycin yield were also explored, revealing the coordination of synthases expression and product biosynthesis in different fermentation stages. Overall, their findings demonstrate the importance of balancing the expression of key limiting factors for efficient secondary biosynthesis and provide valuable reference to in-situ engineering secondary biosynthetic pathways for improved production of other important actinomycete-derived natural products.

Finally, authors conclude that they provided a basis for the overall engineering of ery cluster to further improve erythromycin production, and is also of great significance for the engineering of other secondary biosynthetic pathways to efficiently produce value-added natural products.

==============================

Other comments:

1) Please check for typos throughout the manuscript.

2) With regards to CRISPR-mediated gene editing of microorganisms – authors are kindly encouraged to cite the following article that describes the DNA hydrolytic nuclease activity for a specific microorganism.
DOI: 10.3389/fphar.2018.00114

Author Response

Reviewer 1

Xuemei Zhang and co-authors present a quality and well-written experimental manuscript describing CRISPR/Cas9-mediated multi-loci promoter engineering in ery cluster to improve erythromycin production in Saccharopolyspora erythraea.

Authors fine-tuned the expression of each key limiting ery gene by CRISPR/Cas9-mediated multi-loci promoter engineering. The native promoters were replaced with different heterologous ones of various strengths, generating ten engineered strains, whose erythromycin productions were 2.8- to 6.0-fold improved compared to that of the wild-type strain. Additionally, the optimal expression pattern of multiple rate-limiting genes and preferred engineering strategies of each locus for maximizing erythromycin yield were also summarized. Their work lays a foundation for the overall engineering of ery cluster to further improve erythromycin production. The experience of balancing multiple rate-limiting factors within a cluster is also promising to be applied in other actinomycetes to efficiently produce value-added natural products.

Authors optimized the expression of previously identified six key limiting ery genes, they substituted their native promoters with three different heterologous ones of various strengths using the CRISPR/Cas9-mediated method, and successfully improved erythromycin production of the engineered strains. Moreover, the optimal engineering strategies of each locus for maximizing erythromycin yield were also explored, revealing the coordination of synthases expression and product biosynthesis in different fermentation stages. Overall, their findings demonstrate the importance of balancing the expression of key limiting factors for efficient secondary biosynthesis and provide valuable reference to in-situ engineering secondary biosynthetic pathways for improved production of other important actinomycete-derived natural products.

Finally, authors conclude that they provided a basis for the overall engineering of ery cluster to further improve erythromycin production, and is also of great significance for the engineering of other secondary biosynthetic pathways to efficiently produce value-added natural products.

Other comments:

1) Please check for typos throughout the manuscript.

2) With regards to CRISPR-mediated gene editing of microorganisms – authors are kindly encouraged to cite the following article that describes the DNA hydrolytic nuclease activity for a specific microorganism.

DOI: 10.3389/fphar.2018.00114

Response to Reviewer 1:

  • Thank you very much for raising this point. We have carefully read through the manuscript and corrected the spelling mistakes and grammar errors. These modifications can be viewed by the revision tracking system in the marked revised manuscript “Revised manuscript (microorganisms-2201017)-marked”.
  • Thank you very much for your suggestion. We have read this article titled “Endonuclease from Gram-Negative Bacteria Serratia marcescens Is as Effective as Pulmozyme in the Hydrolysis of DNA in Sputum”, and learned a lot from it. However, the CRISPR technologies introduced in this paper mainly focused on the studies in the actinomycete strains, and the recommended reference did not show strong relevance with this work. Nevertheless, we thank you all the same for your kind suggestion and feel grateful for the knowledge and inspiration that the recommended reference brought for us.

Reviewer 2 Report

The manuscript entitled "CRISPR/Cas9-mediated multi-loci promoter engineering in ery cluster to improve erythromycin production in Saccharineopoly-spora erythraea" aimed to relieve the potential bottlenecks of erythromycin biosynthesis by fine-tuning the expression of each key limiting ery gene by CRISPR/Cas9-mediated multi-loci promoter engineering. This work lays a foundation for the overall engineering of ery cluster to further improve erythromycin production. The experience of balancing multiple rate-limiting factors within a cluster is also promising to be applied in other actinomy-cetes to efficiently produce value-added natural products.

The work is well-conducted, designed and revealed important findings. I have some minor suggestions as follows,

-Add enough paragraph in the discussion section discussing the potential bottlenecks of erythromycin biosynthesis in the view of the present findings,

- It is advisable to re-write the conclusions to highlight the significant findings and clarify how those findings will support the recommended future studies.

- References list should be revised as per the journal guidelines

Author Response

Reviewer 2

The manuscript entitled "CRISPR/Cas9-mediated multi-loci promoter engineering in ery cluster to improve erythromycin production in Saccharineopoly-spora erythraea" aimed to relieve the potential bottlenecks of erythromycin biosynthesis by fine-tuning the expression of each key limiting ery gene by CRISPR/Cas9-mediated multi-loci promoter engineering. This work lays a foundation for the overall engineering of ery cluster to further improve erythromycin production. The experience of balancing multiple rate-limiting factors within a cluster is also promising to be applied in other actinomy-cetes to efficiently produce value-added natural products.

The work is well-conducted, designed and revealed important findings. I have some minor suggestions as follows,

1) Add enough paragraph in the discussion section discussing the potential bottlenecks of erythromycin biosynthesis in the view of the present findings,

2) It is advisable to re-write the conclusions to highlight the significant findings and clarify how those findings will support the recommended future studies.

3) References list should be revised as per the journal guidelines

Response to Reviewer 2:

  • Thank you very much for your valuable suggestion. A paragraph was added at the end of the discussion section as follows: “In addition to the engineering efforts targeting the biosynthetic genes, there are also many successful examples of manipulating genes outside ery cluster to improve erythromycin biosynthesis in erythraea strains. For rational strain engineering, enhancing precursor supply/energy availability [27-29], manipulating regulators/two-component systems [30-32], and overexpressing heterologous non-PKS genes such as the hemoglobin gene vhb [33], are proved to be effective strategies to enhance erythromycin production in S. erythraea to various degrees. The effectiveness of these modifications collectively suggests that erythromycin biosynthesis is probably restricted by a series of complex factors, which are not only located in ery cluster but also distributed among bacterial genome. The comparative genome analysis also revealed the wide distribution of gene mutations of the erythromycin hyper producers compared with the wild-type S. erythraea strain [22,34]. Therefore, to further improve erythromycin production, it is promising to combine the potential beneficial targets in and outside gene cluster together to engineer strain, or express the genetically modified biosynthetic gene cluster in an industrially optimized strain for an alternative [35].”
  • Thank you very much for your valuable suggestion. The conclusion was revised as follows: “To improve erythromycin production, the rational strain engineering focusing on multiple biosynthetic genes is poorly documented compared with the plenty of studies in other aspects. However, synthases encoded by the biosynthetic genes are directly responsible for product formation, which will play dominant roles in the efficient proceedings of secondary biosynthesis. In this study, we fine-tuned the expression levels of six key limiting ery genes by promoter engineering at four different loci, and determined the optimal engineering strategy of each locus using a series of heterologous promoters with a gradient of activities. Ten engineered erythraea strains were constructed for fermentation and transcriptional analysis, and their erythromycin productions were successfully improved by 2.8- to 6.0-fold compared to the wild-type strain. We also demonstrated the appropriate expression of rate-limiting genes, which may significantly vary among targets, is crucial for maximizing erythromycin production. To the best of our knowledge, this work is the first to focus on the expression fine-tuning of four different loci in ery biosynthetic cluster where six key ery genes are involved. This work is strongly complementary to previous engineering efforts in other aspects and provides basis for the overall strain engineering through multiple approaches to further improve erythromycin production. In addition, the attempts to enhance secondary biosynthesis by clarifying the optimal expression pattern of multiple rate-limiting factors with catalytic functions are also applicable to not only in situ engineering but also de novo refactoring of other secondary biosynthetic pathways in actinomycetes, especially for those lacking pathway-specific regulators like ery cluster. Therefore, this work is of great significance for the efficient biosynthesis of other actinomycete-derived value-added natural products.”
  • Thank you very much for raising this point. We have revised the reference list according to the journal guidelines and the modifications can be viewed by the revision tracking system in the marked revised manuscript “Revised manuscript (microorganisms-2201017)-marked”.

Reviewer 3 Report

Review for

 Article 

CRISPR/Cas9-mediated multi-loci promoter engineering in ery cluster to improve erythromycin production in Saccharopolyspora erythraea

Many studies were conducted on

473 document results

TITLE-ABS-KEY ( saccharopolyspora  AND  erythraea ) 

With a special focus on erythromycin, even at a large large scale

Rational design of a 500 m3 fermenter for erythromycin production by Saccharopolyspora erythraea | [用于红霉素生产的 500 m3生物反应器的理性设计] Tan, X., Li, C., Guo, M.    2022      Shengwu Gongcheng Xuebao/Chinese Journal of Biotechnology

38(12), pp. 4692-4704

However, there is still a need for improvements of the production

And this paper is welcome

 Erythromycins are a group of macrolide antibiotics produced by Saccharopolyspora erythraea. Erythromycin biosynthesis, which is a long pathway composed of a series of biochemical reactions, is precisely controlled by the type I polyketide synthases and accessary tailoring enzymes encoded by ery cluster.

 In a previous work, authors have characterized that six genes representing extremely low transcription levels, SACE_0716-SACE_0720 and SACE_0731, played important roles in limiting erythromycin biosynthesis in the wild-type strain S. erythraea NRRL 23338.

 In this new study, to relieve the potential bottlenecks of erythromycin biosynthesis, authors fine-tuned the expression of each key limiting ery gene by CRISPR/Cas9-mediated multi-loci promoter engineering. The native promoters were replaced with different heterologous ones of various strengths, generating ten engineered strains, whose erythromycin productions were 2.8- to 6.0-fold improved compared to that of the wild-type strain.

Why authors chose S. erythraea NRRL 23338 as starting strain?

Generally, secondary biosynthesis is a long biochemical reaction chain where a series of enzymes function orderly and cooperate precisely to synthesize the final product.

As summarized by the authors, the experience of balancing multiple rate-limiting factors within a cluster is also promising to be applied in other actinomycetes to efficiently produce value-added natural products.

The experimental study was well designed, results support the discussion

Figures are nice

Author Response

Reviewer 3

Many studies were conducted on Saccharopolyspora erythraea, with a special focus on erythromycin, even at a large large scale. However, there is still a need for improvements of the production. And this paper is welcome.

Erythromycins are a group of macrolide antibiotics produced by Saccharopolyspora erythraea. Erythromycin biosynthesis, which is a long pathway composed of a series of biochemical reactions, is precisely controlled by the type I polyketide synthases and accessary tailoring enzymes encoded by ery cluster.

In a previous work, authors have characterized that six genes representing extremely low transcription levels, SACE_0716-SACE_0720 and SACE_0731, played important roles in limiting erythromycin biosynthesis in the wild-type strain S. erythraea NRRL 23338.

In this new study, to relieve the potential bottlenecks of erythromycin biosynthesis, authors fine-tuned the expression of each key limiting ery gene by CRISPR/Cas9-mediated multi-loci promoter engineering. The native promoters were replaced with different heterologous ones of various strengths, generating ten engineered strains, whose erythromycin productions were 2.8- to 6.0-fold improved compared to that of the wild-type strain.

Why authors chose S. erythraea NRRL 23338 as starting strain?

Generally, secondary biosynthesis is a long biochemical reaction chain where a series of enzymes function orderly and cooperate precisely to synthesize the final product.

As summarized by the authors, the experience of balancing multiple rate-limiting factors within a cluster is also promising to be applied in other actinomycetes to efficiently produce value-added natural products.

The experimental study was well designed, results support the discussion.

Figures are nice.

Response to Reviewer 3: Thank you very much for your comments. In this study, we chose S. erythraea NRRL 23338 as the starting strain to perform promoter engineering because it is the wild-type erythromycin producer and the genetically tractable model strain. S. erythraea NRRL 23338 is commonly used as the material to study erythromycin biosynthesis mechanism and the parental strain to perform genetic engineering to improve erythromycin production. In addition, S. erythraea NRRL 23338 can also be used as the starting strain for random mutagenesis to obtain high-erythromycin-producing mutants.